# The association of *Helicobacter pylori* infection with serum lipid profiles: An evaluation based on a combination of meta-analysis and a propensity score-based observational approach

**Takeshi Shimamoto[1,2], Nobutake Yamamichi**  **[2]\*, Kenta Gondo[2], Yu Takahashi[2], Chihiro Takeuchi[2], Ryoichi Wada[1], Toru Mitsushima[1], Kazuhiko Koike[2]**

**1** Department of Medical Statistics and Information, Kameda Medical Center Makuhari, Mihama-ku, Chiba, Japan, **2** Department of Gastroenterology, Graduate School of Medicine, The University of Tokyo, Hongo, Bunkyo-ku, Tokyo, Japan

\* nyamamic-tky@umin.ac.jp

## Abstract

### Background

Several previous studies have suggested that *Helicobacter pylori* (*H. pylori*) infection affects the serum lipid profile. However, it remains controversial and the mechanism has not been elucidated. The purpose of this study is to use an epidemiological perspective to evaluate the association between *H. pylori* infection and the serum lipid profile.

### Methods

Multivariate analysis was performed using the data of serum lipid profile, infection status of *H. pylori*, fitness/lifestyle habits, and various subjects' characteristics which were derived from the 15,679 generally healthy individuals in Japan. The average treatment effects (ATEs) of *H. pylori* infection on the serum lipid profile were estimated using augmented inverse probability weighting (AIPW). A meta-analysis was also performed using the 27 studies worldwide in which the status of *H. pylori* infection and at least one serum examination value (high-density lipoprotein cholesterol (HDL-C), low-density lipoprotein cholesterol (LDL-C), total cholesterol (TC), or triglyceride (TG)) were described.

### Results

The ATEs determined with AIPW showed that *H. pylori* infection has significant positive effects on LDL-C and TC (ATE (95% confidence interval [95%CI]) = 3.4 (2.36–4.49) and 1.7 (0.58–2.88), respectively) but has significant negative effects on HDL-C and TG (ATE (95% CI) = −1.2 (−1.74 to −0.72) and −3.5 (−5.92 to −1.06), respectively). The meta-analysis to estimate the association between *H. pylori* infection and the serum lipid profile revealed that *H. pylori* infection is positively associated with LDL-C, TC, and TG (standardized mean

**Data Availability Statement:** All relevant data are within the manuscript and its Supporting Information files.

**Funding:** The author(s) received no specific funding for this work.

**Competing interests:** The authors have declared that no competing interests exist.

**Abbreviations:** Alb, Albumin; ALP, Alkaline Phosphatase; ALT, Alanine Aminotransferase; AST, Aspartate Aminotransferase; BMI, Body Mass Index; DBP, Diastolic Blood pressure; ES, Effect Size; FBG, fasting blood glucose; Hb, hemoglobin; HbA1c, hemoglobin A1c; Hct, hematocrit; HDL-C, high-density lipoprotein cholesterol; LDL-C, low-density lipoprotein cholesterol; MCH, Mean corpuscular hemoglobin; MCHC, mean corpuscular hemoglobin; MCV, Mean Corpuscular Volume; PLT, platelet; RBC, Red Blood Cell; SBP, Systolic Blood Pressure; T-Bil, Total Bilirubin; TC, total cholesterol; TG, triglyceride; TP, Total Protein; UA, uric acid(serum); WBC, White blood cell.

difference [SMD] (95%CI) = 0.11 (0.09–0.12), 0.09 (0.07–0.10) and 0.06 (0.05–0.08), respectively) and negatively associated with HDL-C (SMD = −0.13 (−0.14 to −0.12)).

## Conclusion

Both our multivariate analyses and meta-analysis showed that *H. pylori* infection significantly affects the serum lipid profile, which might lead to various dyslipidemia-induced severe diseases like coronary thrombosis or cerebral infarction.

## Background

More than half of the world's population is presumed to be chronically infected with *Helicobacter pylori* (*H. pylori*) [1]. The prevalence of *H. pylori* is particularly high in East Asian countries such as Japan, China, and South Korea [1,2], and a variety of *H. pylori* impacts on the upper gastrointestinal tract have been widely reported [3,4]. Recently, the effect of *H. pylori* infection on the entire body has drawn considerable attention. For example, several studies have reported an association between *H. pylori* infection and extragastric diseases, such as immune thrombocytopenic purpura, idiopathic sideropenic anemia, and vitamin B12 deficiency [5–7]. Among such extragastric disorders, an effect of *H. pylori* on the lipid profile is one of the most important concerns, especially when considering the very high prevalence of *H. pylori* infection and dyslipidemia all over the world [8]. In recent years, several studies have reported that *H. pylori* infection is associated with the serum lipid profile [9–11], but these findings are considered controversial. Concerning the association between *H. pylori* infection and serum lipid profile, we speculate several mechanisms may be responsible for changes in blood lipid regulation.

One idea is based on the effects of *H. pylori* infection upon the digestive system. A low-grade inflammatory state caused by chronic *H. pylori* infection may interfere with the absorbance of nutrients and could influence the occurrence or evolution of various extragastric diseases. Ghrelin and leptin, both of which are body weight-regulating peptides produced and secreted primarily from the gastric mucosa [12,13], may also play critical roles in this association. Several studies have reported that mucosal atrophy of the stomach induced by *H. pylori* infection greatly affects the homeostasis of leptin and ghrelin [14–18]. These facts indicate that *H. pylori* infection can lead to some appetite related disorders and significant change of body weight. We assume that *H. pylori* infection may cause dysregulated absorption of nutrients in the digestive system, contributing to changes in serum lipids.

The change of lipid profiles may also be due to the effects of the inflammatory response system caused by *H. pylori* infection. Several lines of evidence indicate that the secretion of inflammatory cytokines by cells induced by chronic infection of gram-negative bacteria is related to the change of lipid profiles [19–22]. These investigations indicate that *H. pylori* infection may be involved in the change of lipid profiles through a systemic inflammatory response.

Finally, the effects of *H. pylori* infection may also play a critical role in immune function. It is well established that the eradication of *H. pylori* is effective in treating idiopathic thrombocytopenic purpura (ITP) [23,24]. Furthermore, several studies indicated that a protective effect of *H. pylori* infection against the development of inflammatory bowel disease (IBD) [25]. As ITP and IBD belong to autoimmune diseases, it is possible that *H. pylori* infection may impact the systemic immune system. Furthermore, several studies also suggest that autoimmune disease is associated with the changes in the lipid profile. For example, rheumatoid arthritis (RA),

one of the most common autoimmune diseases, is related to alterations in the lipid profile. The high inflammatory burden of the RA patients was reported to be associated with the low level of high-density lipoprotein cholesterol (HDL-C), low-density lipoprotein cholesterol (LDL-C), and total cholesterol (TC) [26–28]. Thus, we assume that autoimmune abnormality caused by *H. pylori* may have an adverse effect on the serum lipid profile.

For all the aforementioned reasons, we have decided to evaluate the association between *H. pylori* infection status and the serum lipid profiles. Even if the association is not strong, it must be clinically important because both *H. pylori* infection and dyslipidemia are very common disorders, and also because a disordered serum lipid profile can lead to severe life-threatening diseases like coronary thrombosis or cerebral infarction [8]. The prevalence of dyslipidemia has been increasing in Japan according to estimates by the Ministry of Health, Labour and Welfare [29], but a similar trend is observed in many nations and has become a worldwide public health problem [30].

The purpose of this investigation was to evaluate the effects of *H. pylori* infection on the serum lipid profiles based on detailed analyses from an epidemiological perspective.

## Methods

### Study population and ethical approval

This study was approved by the ethics committees of the University of Tokyo, and written informed consent was obtained from each participant before study participation according to the Declaration of Helsinki. The study participants were 19,549 adults with no missing data who underwent a comprehensive medical examination at Kameda Medical Center in Makuhari from January 4 to December 28, 2010. After participants with missing values were omitted, participants with prior gastric surgery (207), taking proton pump inhibitors and/or histamine 2 receptor antagonists (881), having past history of *H. pylori* eradication (1,470), and those taking lipid-lowering drugs (1,312) were further excluded from the investigation, since such confounding factors might adversely affect accurate analysis (Fig 1). The final participants were 15,679 being composed of 8,776 men (mean age 49.6 (9.4) years, range 19–86 years) and 6,903 women (mean age 48.3 (8.9) years, range 20–87 years). In this study, all the participants were asked to respond to the detailed questionnaire (see below), and a serum anti-*H. pylori* IgG antibody test conducted.

### Questionnaires

The Ministry of Health, Labour and Welfare of Japan provided specific health checkups and counseling guidance based on scientific grounds in April 2007, through a program initially started in the fiscal year 2008 [31–33]. We used a part of the questionnaires for fitness and dietary habits included in the medical care system. We asked about fitness habits: "Are you in a habit of doing exercise to sweat lightly for over 30 minutes a time, twice weekly, for over a year?" and "In your daily life, do you walk or do an equivalent amount of physical activity more than one hour a day?". We also surveyed dietary habits: "Is your eating speed quicker than others?", "Do you eat supper 2 hours before bedtime more than three times a week?", "Do you eat snacks after supper more than three times a week?", and "Do you skip breakfast more than three times a week?". We further surveyed weight controls with the question "Have you gained over 10 kg from your weight at age 20?" and "Did you gain or lose over 3 kg during the past year?". In addition to the aforementioned questions, we analyzed answers for two questions as follows: i) "How often do you drink alcohol in a week?" and ii) "Do you have a habit of smoking?". The answers for the question i) were selected from five classifications (never, seldom, sometimes, often, and always), which were further categorized into two groups as

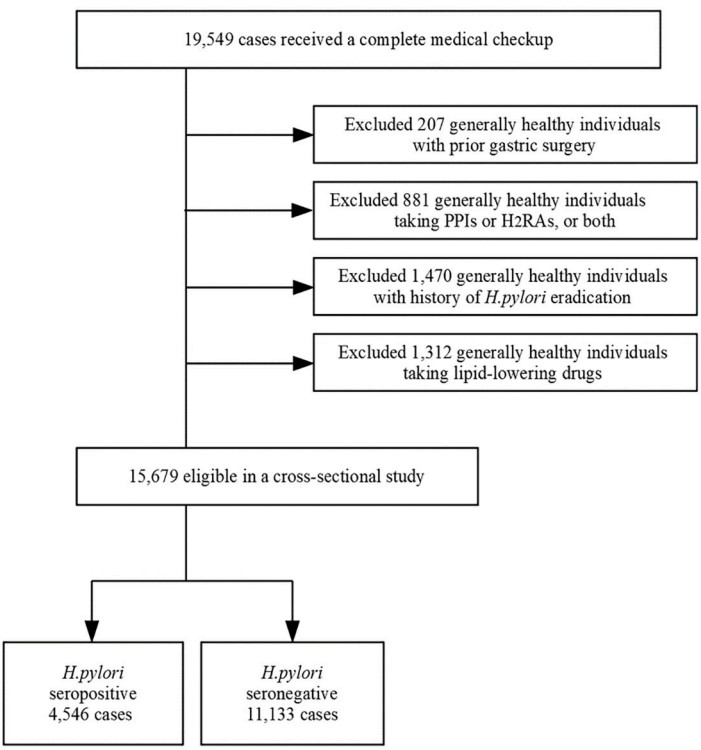

**Fig 1. Study recruitment flowchart.** Of the 19,549 general population participants, we excluded participants with prior gastric surgery (207), those taking proton pump inhibitors and/or histamine 2 receptor antagonists (881), those with a history of *H. pylori* eradication (1,470), and those taking lipid-lowering drugs (1,312). Among the eligible 15,679 participants, the numbers of participants positive and negative for serum *Helicobacter pylori* IgG antibody are shown.

nominal variables: rarely drinking group (never or seldom) and usually drinking group (sometimes, often, or always). The answers for question ii) were categorized into two groups as nominal variables: current or past habitual smoking (smoker group), and lifelong nonsmoking (nonsmoker group). Alcohol intake and smoking status were measured through self-reporting, and a detailed questionnaire including inquiries about past medical history and current medical history was given to all the participants. Answers filled in by the participants were carefully checked by the nursing staff before being recorded in our study database.

## Evaluation of blood chemistry and serum anti-*Helicobacter pylori* antibody

The measurement of serum lipid levels and serum anti-*H. pylori* antibody was performed on fasting blood samples on the day of blood sampling. The measurement of HDL-C and LDL-C was performed on a direct method. The measurement of TG and TC were performed on the free glycerol elimination method and Cholesterol oxidase method, respectively. The serum anti-*H. pylori* antibody was measured using a commercial EIA kit (E-plate "EIKEN" *H. pylori* antibody II, EIKEN Chemical Co Ltd, Tokyo, Japan). According to the manufacture's instruction, an antibody titer above 10 U/ml was considered *H. pylori*-positive.

## Statistical analysis

Using large-scale data from generally healthy individuals, we assessed the current status of the serum lipid profile and *H. pylori* infection in Japan and also evaluated the association between

the two variables. We next conducted a multivariate analysis using all available data to estimate how *H. pylori* infection affects the serum lipid profile. To clarify the effect of *H. pylori* infection on lipid profile, we performed a meta-analysis using all relevant studies published from 1995 to 2016. Finally, we objectively evaluated the estimated results by combining multivariate analysis and meta-analysis. We used JMP 14.2.0, SAS Universal Edition (SAS Institute Inc. Cray, NC, USA) and R statistical package were used for all statistical analyses. The chi-square test was utilized for univariate analysis, multiple logistic regression analysis for multivariate analysis, and an augmented inverse propensity weighted (AIPW) estimator for an estimate of average treatment effects (ATEs).

## Estimating effect sizes and ATEs from cross-sectional data

Cohen's *w* was used for univariate analysis. For multivariate analysis, to estimate the effect of *H. pylori* infection on the serum lipid profile, we used the AIPW with propensity-score (PS) matching was utilized to achieve a better balance between covariates within the matched pairs [34]. The PS was estimated using a logistic regression model that was adjusted to the characteristics of the study participants. Also, to reduce the effect of treatment-selection bias and potential confounding in this cross-sectional study, PS matching was performed by one-to-one pair matching via nearest neighbor matching within a caliper width of 1/5 standard deviation (SD) for the logit of PS without replacement [35]. We used the absolute standardized difference (ASD) to measure covariate balance. An ASD greater than 0.1 represents a meaningful imbalance [36]. The PSs were by defining *H. pylori* infection as a predictive factor and also by defining age, gender, BMI, AST, ALT, ALP, γ-GTP, total bilirubin, TP, Alb, HbA1c, FBG, systolic blood pressure, diastolic blood pressure, UA, RBC, WBC, Hb, Hct, PLT, MCV, MCH, MCHC, smoking, alcohol drinking, fitness habits, and dietary habits as confounding factors. Drinking, dietary habits, and weight controls were treated as yes/no dichotomous variable. With regard to the drinking habits responses, we converted dichotomous variable with "can not drink" and "rarely drink" for "no" and "sometimes", "almost every day" and "every day" for "yes". Other habits and weight controls used the response of yes–no questions. The covariates were selected on the basis of several previous studies [9,37–39].

## Meta-analysis

Meta-analysis was conducted according to the PRISMA guidelines. Previous studies used in our meta-analysis were selected on the basis of the inclusion criteria as follows: RCT, case–control or cohort, cross-sectional design, registration in PubMed, CiNii (Scholarly and Academic Information Navigator) or Ichushi Web (NPO Japan Medical Abstracts Society) databases, and description of HDL-C, LDL-C, TC or TG to statistically evaluation of the association between H. pylori infection and the serum lipid profile. A search was performed for combinations of keywords related to *H. pylori* infection and related to the outcome of interest. Concrete keywords used related to *H. pylori* infection were (an asterisk is a replacement for any ending of the respective term; quotation marks indicate that the term was used as a whole, not each word individually): *pylori**, lipid*, "*H. pylori*," "*Helicobacter pylori*," "lipid profile," "lipid metabolism," obesity, dyslipidemia, and arteriosclerosis. The search in both English and Japanese was completed in November 2016. No restrictions were placed on the language or date of publication when searching the electronic databases. Investigations that showed the results of significance but lacked data on the lipid profile were excluded because we could not calculate the risk difference in the meta-analysis.

To obtain an overview of the association between *H. pylori* infection and the serum lipid profile, we performed meta-analyses using the fixed-effects model. We adopted the definition

of the fixed-effects meta-analysis method because we assumed that the treatment effect was the same for each study. To estimate the risk of bias in our meta-analyses, we investigated publication bias by inspection of funnel plots and we applied the Macaskill's linear regression method test for detecting publication bias of meta-analysis [40].

To the evaluation of the validity of the included studies, we assessed the quality of each study using the Risk of Bias Assessment Tool for Nonrandomized Studies (RoBANS) [41]. The evaluation of risk of bias by RoBANS is as follows: "Selection bias caused by the inadequate selection of participants," "Selection bias caused by the inadequate confirmation and consideration of confounding variable," "Performance bias caused by the inadequate measurement of exposure," "Detection bias caused by the inadequate blinding of outcome assessments," "Attrition bias caused by the inadequate handling of incomplete outcome data," and "Reporting bias caused by the selective reporting of outcomes". We made an overall judgment of these elements by three stages including low, high and unclear.

For the unification of a unit of measure for the lipid profile, first, for results presented as mmol/L, we converted the mean and the SD from mmol/L to mg/dL per the International System of Units. Second, for results presented as the median and interquartile range, when we add the median and the first quartile and the third quartile and divided by 3, we estimated the modified mean. Also, we estimated the SD using the method proposed by Xiang [42].

## Results

### Association between the serum lipid profile and serum anti-*H. pylori* IgG status of the healthy general population in Japan

The levels of the serum lipid profile were compared between the participants seropositive and seronegative for *H. pylori* IgG (Table 1). The *P*-value was calculated by Welch's *t*-test, and

**Table 1. Characteristics of serum HDL-C, LDL-C, TC, and TG categorized on the basis of the status of serum *Helicobacter pylori* IgG antibody and age groups.**

| | HDL-C | | LDL-C | | TC | | TG | | |
|---|---|---|---|---|---|---|---|---|---|
| | seropositive | seronegative | seropositive | seronegative | seropositive | seronegative | seropositive | seronegative | |
| | N = 4546 | N = 11133 | N = 4546 | N = 11133 | N = 4546 | N = 11133 | N = 4546 | N = 11133 | |
| Age (year) | | | | | | | | | |
| N (Total/male/female) | | | | | | | | | |
| <40 | 62.9 (15.1) | 65.3 (16.3) | 114.3 (30.1) | 109.2 (29.4) | 189.2 (31.3) | 185.8 (30.6) | 92.1 (61.5) | 88.1 (57.4) | |
| N (2539/1328/1211) | | | | | | | | | |
| 40–49 | 63.6 (15.9) | 65.8 (17.1) | 122.6 (31.2) | 119.0 (31.1) | 199.1 (33.6) | 197.7 (31.7) | 103.3 (81.3) | 102.4 (78.9) | |
| N (5685/3068/2617) | | | | | | | | | |
| 50–59 | 63.3 (16.4) | 66.4 (17.5) | 132.4 (29.0) | 128.9 (29.5) | 209.6 (31.1) | 209.1 (31.3) | 111.6 (82.0) | 110.9 (75.9) | |
| N (5385/3058/2327) | | | | | | | | | |
| 60≦ | 63.2 (16.3) | 65.7 (16.8) | 133.8 (28.9) | 130.7 (28.9) | 211.2 (31.7) | 210.4 (30.6) | 110.8 (72.2) | 108.1 (67.3) | |
| N (2070/1322/748) | | | | | | | | | |
| all ages | 63.3 (16.1) | 65.9 (17.1) | 128.8 (30.3) | 121.2 (31.0) | 205.6 (32.6) | 200.0 (32.5) | 107.7 (78.4) | 102.7 (73.6) | |
| N (15679/8776/6903) | | | | | | | | | |
| *t*-value | 8.9 | | 14.1 | | 9.8 | | 3.7 | | |
| df | 8886.7 | | 8629.6 | | 8406.3 | | 7988.6 | | |
| *P*-value | <0.001 | | <0.001 | | <0.001 | | <0.001 | | |
| effect size | 0.15 | | 0.25 | | 0.17 | | 0.07 | | |

The *P*-value was calculated by Welch's *t*-test. Effect size was calculated by Cohen's *d*. A two-tailed *P*-value less than 0.05 was considered statistically significant. Numbers in parentheses show the SD of the mean.

effect sizes (ESs) were calculated by Cohen's *d*. The participants with chronic *H. pylori* infection had lower values for HDL-C and higher values for LDL-C, TC, and TG. The associations of the four types of lipid with serum *H. pylori* IgG were all statistically significant(HDL-C: *t* (8886.7) = 8.9, *P* < 0.001; LDL-C: *t* (8629.6) = 14.1, *P* < 0.001; TC: *t* (8406.3) = 9.8, *P* < 0.001; TG: *t* (7988.6) = 3.7, *P* < 0.001). Judging from the value of ES, *H. pylori* infection had a substantial effect on LDL-C (ES = 0.25). Conversely, *H. pylori* infection had only small effects on HDL-C, TC, and TG (ES = 0.15, 0.17 and 0.07 respectively).

Concerning gender, *H. pylori*-positive males were significantly associated with HDL-C, LDL-C, and TC; namely men with chronic *H. pylori* infection had lower values for HDL-C and higher values for LDL-C, and TC (S1 Table). Men with chronic *H. pylori* infection had lower values of TG compared with men without it, though the difference between the two was not statistically significant. On the other hand, *H. pylori*-positive females were significantly associated with all types of lipids; namely, women with chronic *H. pylori* infection had lower values for HDL-C and higher values for LDL-C, TC, and TG (S2 Table). Men had lower values for HDL-C and TC and higher values for LDL-C and TG compared with women, and there were statistically significant differences between the sexes in all types of lipid (HDL-C: *t* (14304) = 56.1, *P* < 0.001; LDL-C: *t* (14374) = 13.1, *P* < 0.001; TC: *t* (14317) = 3.8, *P* = 0.002; TG: *t* (14666) = 42.5, *P* < 0.001). Interestingly, chronic *H. pylori* infection had a large effect on LDL-C and TC for females. The ES of women with chronic *H. pylori* infection in LDL-C was the largest (ES = 0.38) out of all types of lipids (S2 Table). Furthermore, women with chronic *H. pylori* infection showed a statistical and clinical difference at 0.2, 0.29, and 0.15 in HDL-C, TC, and TG, respectively (S2 Table). In contrast, men with chronic *H. pylori* infection showed a statistical and clinical difference at 0.13 only in LDL-C out of all types of lipids (S1 Table).

## Checking the balance of confounding factors in logistic regression using the standardized difference

The predicted probabilities of *H. pylori*-positivity were calculated via a logistic regression model, using carefully selected covariates as shown in S3 Table. Of the possible 11,133 *H. pylori*-negative participants, 4376 were matched with *H. pylori*-positive participants. Age (ASD:0.614), ALP (ASD:0.191), Alb (ASD:0.185), HbA1c (ASD:0.139), systolic BP (ASD:0.157), diastolic BP (ASD:0.169), UA (ASD:0.107), WBC (ASD:0.127), and smoking habit (ASD:0.127) were considered poorly balanced before matching. However, all the covariates in the estimation of HDL-C, LDL-C, TC, and TG were considered well balanced after PS matching judging from the values of ASD. The covariate balance in the matched population was improved by a matching method; the effect of selection bias and potential confounding in our cross-sectional study was successfully reduced.

## Estimation of ATEs and confidence intervals concerning the relationship between *H. pylori* infection and serum lipid profile using AIPW

All the ATEs were estimated using AIPW (Table 2). Our data showed that *H. pylori* infection had positive effects on LDL-C and TC with statistical significance (ATE: 3.4 and 1.7, respectively). In contrast, *H. pylori* infection had negative effects on HDL-C and TG with statistical significance (ATE: −1.2 and −3.5, respectively).

## Assessment of the risk of bias of individual studies for performing a meta-analysis

The 33 case–control studies, including our data, were evaluated, and we found that 28 studies fulfilled the inclusion criteria (S4 Table) and 5 studies did not (S5 Table). Consequently, data

**Table 2. Summary of augmented inverse probability weighting reflecting the effect of *Helicobacter pylori* infection on the serum lipid profile in generally healthy individuals.**

|  | *Helicobacter pylori* infection | Estimate | SE | 95% CI | z-test | P-value |
|---|---|---|---|---|---|---|
| HDL-C | seronegative | 65.5 | 0.16 | 64.14–65.79 | 397.5 | <0.001 |
|  | seropositive | 64.2 | 0.23 | 63.78–64.69 | 279.1 | <0.001 |
|  | ATE | -1.2 | 0.26 | (-1.74)–(-0.72) | -4.7 | <0.001 |
| LDL-C | seronegative | 122.3 | 0.30 | 121.72–122.89 | 409.4 | <0.001 |
|  | seropositive | 125.7 | 0.48 | 124.79–126.67 | 261.8 | <0.001 |
|  | ATE | 3.4 | 0.54 | 2.36–4.49 | 6.3 | <0.001 |
| TC | seronegative | 201.1 | 0.31 | 200.45–201.65 | 654.0 | <0.001 |
|  | seropositive | 202.8 | 0.52 | 201.76–203.81 | 387.3 | <0.001 |
|  | ATE | 1.7 | 0.58 | 0.58–2.88 | 3.0 | 0.003 |
| TG | seronegative | 105.2 | 0.86 | 103.46–106.83 | 122.6 | <0.001 |
|  | seropositive | 101.7 | 1.00 | 99.69–103.62 | 101.4 | <0.001 |
|  | ATE | -3.5 | 1.24 | (-5.92)–(-1.06) | -2.8 | 0.007 |

The average treatment effect (ATE) shows the effects of *H. pylori* infection on lipid profiles. Seronegative: the augmented inverse probability weighting of response in the absence of *H. pylori*, Seropositive: the augmented inverse probability weighting of response in the presence of *H. pylori*. ATE would be the sample average of seropositive minus seronegative. HDL-C: high-density lipoprotein cholesterol, LDL-C: low-density lipoprotein cholesterol, TC: total cholesterol, TG: triglyceride, $\chi^2$: the chi-square test statistic, SE: standard error, 95% CI: 95% confidence interval. The *P*-value used for ATE, a two-tailed *P*-value less than 0.05 was considered statistically significant.

from the 28 studies (67,290 *H. pylori*-positive and 53,859 *H. pylori*-negative) were used in the meta-analysis (S1 Fig).

S2 Fig shows an assessment of the validity of the studies included. We judged the blinding of participants (selection bias), confounding variables (selection bias), and measurement of exposure (performance bias) in several studies as high risk of bias. Conversely, we judged the blinding of outcome assessments (detection bias) as low risk of bias. Furthermore, we judged incomplete outcome data (attrition bias) and selective outcome reporting (reporting bias) as unclear risk of bias. Although exposure measurement is not blinded in every study, we judged a considerably low probability of measurement bias because lipid profile and the diagnosis of *H. pylori* are the mere screening blood tests conducted in clinics and health checkups.

### Estimation of the pooled mean ESs and confidence intervals in meta-analyses to examine the association between the serum lipid profiles and *H. pylori* infection

Fig 2 shows the results of the meta-analyses. All results were presented graphically in forest plots, and the diamonds at the bottom show the pooled risk differences for all studies with the 95% confidence interval. As shown in Fig 2A, the meta-analysis of 27 studies showed a significant negative association between *H. pylori* infection and serum HDL-C (standardized mean difference [SMD], −0.13; 95% CI, −0.14 to −0.12; *P* for heterogeneity, <0.01). However, as shown in Fig 2B, the meta-analysis of 22 studies showed a significant positive association between *H. pylori* infection and serum LDL-C (SMD, 0.11; 95% CI, 0.09 to 0.12; *P* for heterogeneity, <0.01). Similarly, as shown in Fig 2C, the meta-analysis of 27 studies showed a significant positive association between *H. pylori* infection and serum TC (SMD, 0.08; 95% CI, 0.07 to 0.09; *P* for heterogeneity, <0.01). Further, as shown in Fig 2D, the meta-analysis of 25 studies showed a significant positive association between *H. pylori* infection and serum TG (SMD, 0.06; 95% CI, 0.05 to 0.08; *P* for heterogeneity, <0.01). Altogether, our meta-analyses showed a

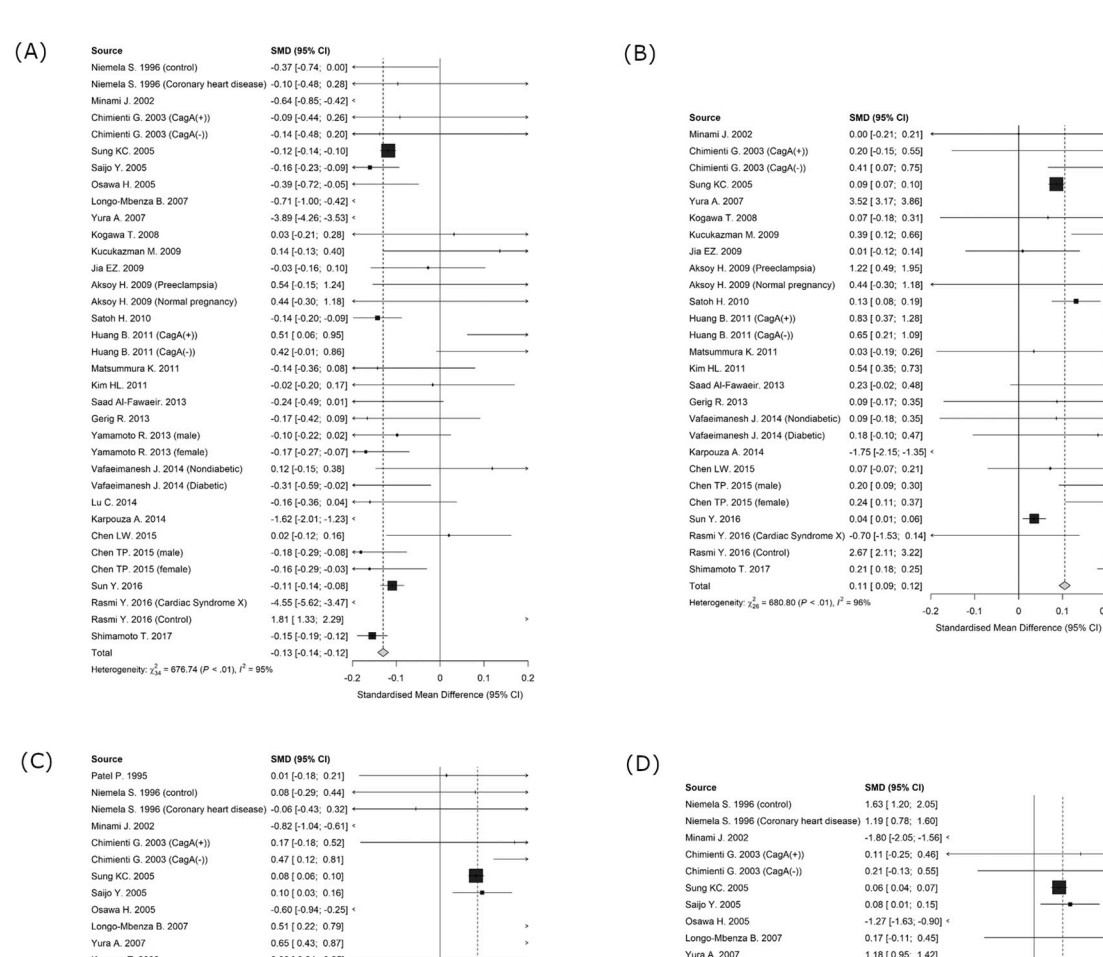

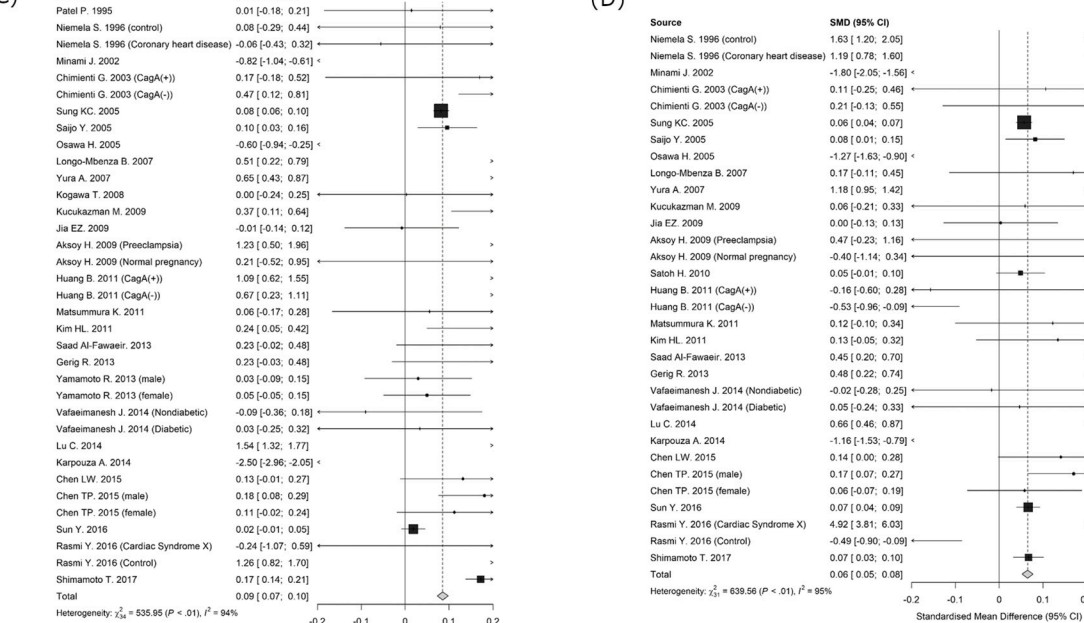

**Fig 2. Forest plots of odds ratio (OR) with 95% confidence interval (CI) showing the effect of *Helicobacter pylori* infection in each lipid profile.** The gray box represents an estimate of the OR in the respective studies, and the horizontal line indicates the 95% CI for the respective studies. Diamonds at the bottom represent the pooled estimate of OR. Weights are from fixed-effects meta-analysis. The chi-squared ($x^2$) test revealed the presence of heterogeneity. The $I^2$ value shows the extent of heterogeneity. (A) High-density lipoprotein cholesterol (HDL-C), (B) low-density lipoprotein cholesterol (LDL-C), (C) total cholesterol (TC), and (D) triglyceride (TG).

significant association of *H. pylori* infection with the serum lipid profile. In addition, we inspected funnel plots to check the existence of publication bias (S3 Fig). Though asymmetry or small-study effects were detected on all the lipid profiles, no statistically unacceptable results were observed in the tests for publication bias by Macaskill's linear regression method (HDL-C: $F(1, 33) = 0.085$, $P = 0.773$, LDL-C: $F(1, 25) = 0.243$, $P = 0.627$, TC: $F(1, 30) = 0.042$, $P = 0.840$, TG: $F(1, 30) = 0.058$, $P = 0.811$).

## Discussion

The results of our univariate analyses revealed that *H. pylori* infection has a statistical and clinical relevant relationship with the serum lipid profile (Table 1). Multivariate analyses with AIPW indicated that *H. pylori* infection has significant positive effects on LDL-C and TC and negative effects on HDL-C and TG (Table 2). The possibility of an outlier impacting these results is small because we excluded the participants who were taking lipid-lowering drugs in both the univariate and multivariate analyses. The meta-analysis indicated that *H. pylori* infection is positively associated with LDL-C, TC, and TG and negatively associated with HDL-C with statistical significance. In our meta-analyses, publication bias is an ignorable matter, judging from the results of Macaskill's linear regression method. In addition, the 95% confidence interval overlap in many studies and the estimated effect are in the same direction.

All the results were consistent with that *H. pylori* infection can have a statistically significant effect on the serum lipid profile. Judging from the certainty of the evidence, it is strongly suggested that there is a causal relationship between them. Although interpreting the contradictory association of TG with *H. pylori* infection is difficult, it may be due to the fact that the study participants were generally healthy subjects (free from severe diseases) who underwent annual health check-up. In total, we are now convinced that chronic *H. pylori* infection can lead to increased levels of serum LDL-C and TC and also can lead to a reduced level of serum HDL-C.

These results are noteworthy because both high levels of serum LDL-C and low levels of serum HDL-C are established risk factors of atherosclerosis and coronary artery disease [43]. The rate of *H. pylori* infection has been gradually decreased in Japan, but a similar trend for *H. pylori* infection is observed all over the world. Therefore, it is important to investigate the changing prevalence of *H. pylori* infection along with the serum lipid profile. Results from the present investigation suggest that *H. pylori* infection affects the serum lipid profile and can indirectly influence the various diseases caused by abnormal lipid metabolism. This finding must be clinically important since both *H. pylori* infection and dyslipidemia are very common chronic disorders.

There are some experimental considerations that may have limited this investigation. Because of the cross-sectional study design of this investigation, we were unable to perform accurate analyses of the causal effects. Study participants also completed a comprehensive medical examination, and we could not evaluate the actual conditions of the patients with severe health problems or critical diseases. Also, the serum anti-*H. pylori* antibody test was used to diagnose the presence of *H. pylori*, but it is inferior to a urea breath test or histopathological examination in the quality of infection diagnosis. Finally, there is a limited number of available studies in this field, which could have resulted in a lower-quality of meta-analyses.

## Conclusions

Both multivariate analysis using the large-scale data for generally healthy subjects in Japan and meta-analysis based on the previous studies worldwide showed that *H. pylori* infection significantly impacts the serum lipid profile of healthy humans. Since *H. pylori* infection and

dyslipidemia are common disorders worldwide, the significant association between the two is kindly to have clinical utility.

## Supporting information

**S1 Table. Distributions of the values of HDL-C, LDL-C, TC, and TG categorized based on the status of serum Helicobacter pylori IgG antibody and age groups of males.**
(XLSX)

**S2 Table. Distributions of the values of HDL-C, LDL-C, TC, and TG categorized based on the status of serum Helicobacter pylori IgG antibody and age groups of females.**
(XLSX)

**S3 Table. Estimation of the absolute standardized mean differences before and after matching the respective covariates to compare Helicobacter pylori seropositivity and sero-negativity in generally healthy individuals.**
(XLSX)

**S4 Table. A summary of the characteristics of cohort and case-control studies included in a meta-analysis, which was performed to compare the association of Helicobacter pylori to lipid profile.**
(XLSX)

**S5 Table. A summary of the characteristics of cohort and case-control studies excluded in a meta-analysis, which was performed to compare the association of Helicobacter pylori to lipid profile.**
(XLSX)

**S1 Fig. PRISMA flow diagram.**
(TIFF)

**S2 Fig. The risk of bias summary and graph by the authors' judgments.** (A) The risk of bias summary by authors' judgments about each bias element for each study. (B) The risk of bias graph by assessment of the risk of bias across studies. Bias is assessed as judgment (high, low, or unclear) for individual elements. The diagonal stripes show a low risk of bias, pin dot shows unclear risk of bias and black fill shows a high risk of bias.
(TIFF)

**S3 Fig. Publication bias.**
(TIFF)

**S1 Checklist.**
(DOCX)

**S1 Dataset.**
(ZIP)

## Acknowledgments

We thank Kameda Medical Center Makuhari staffers for great assistance with establishment and maintenance of the study database.

## Author Contributions

**Conceptualization:** Takeshi Shimamoto.

**Data curation:** Takeshi Shimamoto.

**Formal analysis:** Takeshi Shimamoto.

**Investigation:** Takeshi Shimamoto.

**Methodology:** Takeshi Shimamoto.

**Project administration:** Nobutake Yamamichi.

**Software:** Takeshi Shimamoto.

**Supervision:** Nobutake Yamamichi.

**Validation:** Takeshi Shimamoto, Kenta Gondo.

**Writing – original draft:** Takeshi Shimamoto.

**Writing – review & editing:** Nobutake Yamamichi, Kenta Gondo, Yu Takahashi, Chihiro Takeuchi, Ryoichi Wada, Toru Mitsushima, Kazuhiko Koike.

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
