## [Decision Letter · Decision Letter 0]

6 Apr 2020

PONE-D-19-35830

The association of Helicobacter pylori infection with serum lipid

profiles: Evaluation based on a combination of systematic review and meta-analysis, along with the average treatment effects using a propensity score-based observational approach

PLOS ONE

Dear Dr. Yamamichi,

Thank you for submitting your manuscript to PLOS ONE. After careful consideration, we feel that it has merit but does not fully meet PLOS ONE’s publication criteria as it currently stands. Therefore, we invite you to submit a revised version of the manuscript that addresses the points raised during the review process.

The Authors are invited to address all reviewer's comments

We would appreciate receiving your revised manuscript by May 30, 2020. To enhance the reproducibility of your results, we recommend that if applicable you deposit your laboratory protocols in protocols.io, where a protocol can be assigned its own identifier (DOI) such that it can be cited independently in the future. For instructions see: http://journals.plos.org/plosone/s/submission-guidelines#loc-laboratory-protocols

We look forward to receiving your revised manuscript.

Kind regards,

Paolo Magni

Academic Editor

PLOS ONE

Journal Requirements:

Reviewers' comments:

Reviewer's Responses to Questions

**Comments to the Author**

1. Is the manuscript technically sound, and do the data support the conclusions?

Reviewer #1: Yes

2. Has the statistical analysis been performed appropriately and rigorously? 

Reviewer #1: Yes

3. Have the authors made all data underlying the findings in their manuscript fully available?

Reviewer #1: Yes

4. Is the manuscript presented in an intelligible fashion and written in standard English?

Reviewer #1: Yes

5. Review Comments to the Author

Reviewer #1: The paper by Shimamoto et al describes a combination of a clinical study and a meta-analysis on the association between H. pylori infection and plasma lipid profile. The study is significant, relatively novel and well executed. There are only few minor issues in this study that require further clarification.

Specific comments:

1. May I suggest a much shorter title

2. anti-H. pylori antibody titre – is there any proven relationship between the titre and severity of infection, either author’s own data or finding by others (in the latter case, please provide a reference)?

3. Please provide details on how the plasma lipids were measured

4. Was duration of the infection taken into consideration?

5. p. 15 Discussion: “H. pylori infection has a significant effect on the serum lipid profile”. I suggest to be more careful in suggesting causation. This is a clinical study discovering associations and you can’t exclude a possibility that plasma lipid levels have an effect on probability and severity of the infection or both have a common cause.

6. p.16 Discussion “study participants were healthy individuals” – what is the definition of healthy if they carry an infection?

6. PLOS authors have the option to publish the peer review history of their article (what does this mean?). If published, this will include your full peer review and any attached files.

Reviewer #1: No

---

## [Author Response · Author response to Decision Letter 0]

18 May 2020

< Editor’s comments>

http://www.journals.plos.org/plosone/s/file?id=wjVg/PLOSOne_formatting_sample_main_body.pdf and http://www.journals.plos.org/plosone/s/file?id=ba62/PLOSOne_formatting_sample_title_authors_affiliations.pdf.

 As the editor pointed out, in accordance with the manuscript body formatting guidelines of PloS ONE and we modified the whole text, especially Abstract and Reference.

 In specific terms, we used the medical care system specific to Japan to evaluate physical activity and dietary habits such as walking, the number of meals and mealtime. We provided sufficient details in the manuscript (On page 7, line 17). Additionally, we show the translation of the questionnaire on specific health examination and their content as reference material, in the following reference materials. We listed them in reference numbers 32 and 33.

Specific Health Checkups and Specific Health Guidance (reference number 32)

https://www.mhlw.go.jp/english/wp/wp-hw3/dl/2-007.pdf

Questionnaire on specific health examination in English (reference number 33)

http://eng.amda-imic.com/oldpage/amdact/PDF/eng/spe-he-ex-e.pdf

As the editor pointed out, in accordance with the sharing data publicly of PloS ONE and we prepared the minimal anonymized data set.

 As the editor pointed out, we amend the title.

The previous title: The association of Helicobacter pylori infection with serum lipid profiles: Evaluation　based on a combination of systematic review and meta-analysis, along with the effect size using a propensity　score-based observational approach

The current title: The association of Helicobacter pylori infection with serum lipid profiles: an evaluation　based on a combination of meta-analysis and a propensity score-based observational approach

As the editor pointed out, we described them in the new text (on page 17, line 11).

< Reviewer’s comments>

1. May I suggest a much shorter title.

 As per the reviewer’s suggestion, we shortened a title. (On page 1, line 1).

The previous title: The association of Helicobacter pylori infection with serum lipid profiles: Evaluation based on a combination of systematic review and meta-analysis, along with the effect size using a propensity score-based observational approach

The current title: The association of Helicobacter pylori infection with serum lipid profiles: an evaluation based on a combination of meta-analysis and a propensity score-based observational approach

2. anti-H. pylori antibody titre - is there any proven relationship between the titre and severity of infection, either author's own data or finding by others (in the latter case, please provide a reference)?

　If we consider atrophic gastritis as the severity of infection, we have reported that the association of anti-H. pylori antibody titer with the diagnosis of atrophic gastritis (Yamamichi et al., 2016).

For details of the paper, please refer to the following Web page of the journal Gastric Cancer. 

https://link.springer.com/article/10.1007/s10120-015-0515-y

To provide a simple explanation, as shown in Fig. 3 in this paper, our evidence indicates that the anti-H. pylori antibody titer of the presence of atrophic gastritis was considerably more expensive than the absence of atrophic gastritis, although there is no hard biological evidence at this time to support this hypothesis.

3. Please provide details on how the plasma lipids were measured

As the reviewer pointed out, we described them in the new text (on page 8, line 17). 

4. Was duration of the infection taken into consideration?

 As the reviewer pointed out, we consider the issue of the duration of the infection is the subject for further study. However, the timing of infection establishment of H. pylori is very difficult because the infection of H. pylori is more common in their early childhood with weak immune systems. There is inadequate evidence to prove that it is possible to pseudo-fix a reasonable time for the duration of the infection, for example, three years old. Additionally, the route of contamination of H. pylori remains to be completely elucidated. Therefore, in this paper, it is difficult to consider the duration of the infection.

5. p. 15 Discussion: "H. pylori infection has a significant effect on the serum lipid profile". I suggest to be more careful in suggesting causation. This is a clinical study discovering associations and you can't exclude a possibility that plasma lipid levels have an effect on probability and severity of the infection or both have a common cause.

As per the reviewer’s suggestion, we changed the contestation from a description of unquestionable evidence to a description of potential evidence. However, from examining the analyses, we are now convinced that chronic H. pylori infection can lead to increased levels of serum LDL-C and TC and also can lead to a reduced level of serum HDL-C. (on page 16 , line 4).

6. p.16 Discussion "study participants were healthy individuals" - what is the definition of healthy if they carry an infection?

 Healthy individuals of this study are characterized by without a history of severe past and present disease and have not any subjective symptoms. In other words, the subjects who underwent a comprehensive medical examination are generally healthy subjects. As the reviewer pointed out, we described a rich description of generally healthy subjects in the new text. Additionally, we changed a description of the subjects from "healthy individual" to "generally healthy subject" and defined. (on page 16, line 7).

---

## [Decision Letter · Decision Letter 1]

27 May 2020

The association of Helicobacter pylori infection with serum lipid profiles: an evaluation based on a combination of meta-analysis and a propensity score-based observational approach

PONE-D-19-35830R1

Dear Dr. Nobutake Yamamichi,

We are pleased to inform you that your manuscript has been judged scientifically suitable for publication and will be formally accepted for publication once it complies with all outstanding technical requirements.

With kind regards,

Paolo Magni

Academic Editor

PLOS ONE

Additional Editor Comments (optional):

Reviewers' comments:

Reviewer's Responses to Questions

**Comments to the Author**

1. If the authors have adequately addressed your comments raised in a previous round of review and you feel that this manuscript is now acceptable for publication, you may indicate that here to bypass the “Comments to the Author” section, enter your conflict of interest statement in the “Confidential to Editor” section, and submit your "Accept" recommendation.

Reviewer #1: All comments have been addressed

2. Is the manuscript technically sound, and do the data support the conclusions?

Reviewer #1: Yes

3. Has the statistical analysis been performed appropriately and rigorously? 

Reviewer #1: Yes

4. Have the authors made all data underlying the findings in their manuscript fully available?

Reviewer #1: Yes

5. Is the manuscript presented in an intelligible fashion and written in standard English?

Reviewer #1: Yes

6. Review Comments to the Author

Reviewer #1: (No Response)

7. PLOS authors have the option to publish the peer review history of their article (what does this mean?). If published, this will include your full peer review and any attached files.

Reviewer #1: No

---

## [Editor Report · Acceptance letter]

29 May 2020

PONE-D-19-35830R1 

The association of Helicobacter pylori infection with serum lipid profiles: an evaluation based on a combination of meta-analysis and a propensity score-based observational approach 

Dear Dr. Yamamichi:

I am pleased to inform you that your manuscript has been deemed suitable for publication in PLOS ONE. Congratulations! Your manuscript is now with our production department. 

With kind regards,

on behalf of

Prof. Paolo Magni 

Academic Editor

PLOS ONE